# Bacteriophage-Encoded DNA Polymerases—Beyond the Traditional View of Polymerase Activities

**DOI:** 10.3390/ijms23020635

**Published:** 2022-01-07

**Authors:** Joanna Morcinek-Orłowska, Karolina Zdrojewska, Alicja Węgrzyn

**Affiliations:** 1Department of Molecular Biology, Faculty of Biology, University of Gdansk, Wita Stwosza 59, 80-308 Gdansk, Poland; joanna.morcinek-orlowska@phdstud.ug.edu.pl (J.M.-O.); karolina.zdrojewska@phdstud.ug.edu.pl (K.Z.); 2Laboratory of Phage Therapy, Institute of Biochemistry and Biophysics, Polish Academy of Sciences, Kładki 24, 80-822 Gdansk, Poland

**Keywords:** bacteriophage, DNA polymerases, DNA replication

## Abstract

DNA polymerases are enzymes capable of synthesizing DNA. They are involved in replication of genomes of all cellular organisms as well as in processes of DNA repair and genetic recombination. However, DNA polymerases can also be encoded by viruses, including bacteriophages, and such enzymes are involved in viral DNA replication. DNA synthesizing enzymes are grouped in several families according to their structures and functions. Nevertheless, there are examples of bacteriophage-encoded DNA polymerases which are significantly different from other known enzymes capable of catalyzing synthesis of DNA. These differences are both structural and functional, indicating a huge biodiversity of bacteriophages and specific properties of their enzymes which had to evolve under certain conditions, selecting unusual properties of the enzymes which are nonetheless crucial for survival of these viruses, propagating as special kinds of obligatory parasites. In this review, we present a brief overview on DNA polymerases, and then we discuss unusual properties of different bacteriophage-encoded enzymes, such as those able to initiate DNA synthesis using the protein-priming mechanisms or even start this process without any primer, as well as able to incorporate untypical nucleotides. Apart from being extremely interesting examples of biochemical biodiversity, bacteriophage-encoded DNA polymerases can also be useful tools in genetic engineering and biotechnology.

## 1. Introduction

In 1953, Watson and Crick resolved the structure of double-stranded DNA. This discovery enabled a huge development in molecular biology. Subsequent work on understanding the cellular mechanisms of DNA replication revealed that this process is tightly regulated and requires the joint action of several protein factors that interact with themselves and with DNA. The key enzyme in DNA replication is DNA polymerase, responsible for the processivity and fidelity of DNA synthesis [1,2].

Mechanistically, five main steps can be distinguished when describing DNA synthesis by polymerase. The first is the physical formation of hydrogen bonds between the relevant bases: two bonds between adenine and thymine or three bonds between cytosine and guanine. Subsequently, water is removed from the active site of the polymerase. As a result, the nucleotide can be attached by changing the conformation of the enzyme in the active site. This results in the formation of a phosphodiester bond between the last base of the primer and the new nucleotide. The free 3′-OH end is required for this [3,4].

Polymerases are common across all taxa since every organism needs to replicate its genetic material. There is no exception with viruses—after infecting the host cell they have to rapidly copy their genome and produce progeny particles. Because the genome of different viral taxa can be either DNA or RNA of different types (linear, circular, single-or double-stranded), viruses rely on different polymerases to meet their replication needs. Logically, DNA-dependent DNA polymerases are used to replicate the genome of DNA viruses [5].

A specific kind of virus is the bacteriophage, which infects bacteria. The ability to produce phage progeny particles often depends on proteins encoded by host genes, including host DNA polymerase. However, there are some bacteriophages that encode their own polymerases. Among the best known DNA polymerase-encoding viruses, there are ϕ29, T7, T4 and T4-like RB69 phages [6,7,8,9], but other bacteriophages coding for their own DNA polymerases are being constantly discovered.

Due to their prevalence across the wide variety of environmental conditions, bacteriophages are a source of exploration for yet undiscovered DNA polymerases. They may have unusual properties that exceed far beyond the traditional notion of DNA polymerase, that is an enzyme of specific structure with obligatory 5′ to 3′ DNA synthesis activity and optional exonuclease activity [2,5]. In this review we highlight the basic information about ‘typical’ DNA polymerases of the best-known bacteriophages and show the recent examples of phage-encoded polymerases with noncanonical activities or specific, previously undescribed features.

## 2. DNA Polymerases

DNA polymerases have a structure resembling the right hand. They have three main domains: palm, fingers, and thumb. The fingers domain interacts with incoming ssDNA and dNTPs, and the thumb domain binds dsDNA. The palm domain contains an active site where amino acid residues bind divalent ions, allowing primer elongation on the DNA template [10].

Despite their structural similarities, DNA polymerases, based on phylogenetic analysis and nucleotide sequence similarity, were divided into seven families: A, B, C, D, X, Y and RT (Table 1) [11]. The DNA polymerases of family A have replicative and repair activities. With the 3′→5′ exonuclease activity, proofreading is provided, allowing the repair of a mispaired nucleotide, while the 5′→3′ exonuclease activity allows the removal of RNA primers. Polymerases of this family are found in eukaryotic organisms (Pol γ, Pol θ, Pol v), bacteria (Pol I) and viruses (T7 DNA pol) [4,12,13].

Family B polymerases, unlike family A enzymes, lack 5′→3′ exonucleolytic activity. In the case of polymerases found in eukaryotic cells, most of them also lack the domain responsible for 3′→5′ exonuclease activity. However, in other taxa, 3′→5′ exonuclease has a 1000-fold higher activity than the Klenow fragment of the *Escherichia coli* Pol I. B family polymerases are found in eukaryota (Pol ζ, α, σ), bacteria (Pol II), archea (DNA pol B) and viruses (T4 DNA pol) [4,6,12,13,14,15].

The C polymerase family comprises enzymes that are the main proteins responsible for chromosomal replication. They do not show much similarity to the polymerases of the A and B families. They are holoenzymes whose activity depends on interactions with at least 10 other proteins. These enzymes are found in bacteria, and an example of such a polymerase is Pol III [12,13,16].

The D family enzymes are characterized by the presence of two subunits: the small DP1 (responsible for 3′→5′ exonuclease activity) and the large DP2, which has a polymerase activity. To date, they have only been discovered in euryarchaeota, and an example is Pol D [13].

Polymerases included in the X family are small and monomeric. They catalyze replication, are involved in recombination and DNA repair, and have DNA repair capabilities that are mainly based on gap filling and joining broken DNA strands. These enzymes can be found in many taxa: eukaryota (Pol β, σ, λ, μ), bacteria (Pol X), archea (Pol X), and viruses (African Swine Fever Virus DNA pol) [4,13,17].

In the Y family, polymerases have an additional little finger domain that allows them to bypass strand breaks. As a result, even damaged DNA can be replicated. Examples of such enzymes are the polymerases found in eukaryota (Pol ι, κ, η), bacteria (Pol IV and V) and archea (Dpo4 DNA pol) [4,13,18,19].

There are also DNA polymerases that are RNA-dependent. They enable the synthesis of DNA strands on the ssRNA template. They belong to the RT family. The process of reverse transcription is also used by telomerase in the elongation of telomeres in eukaryotic cells. Another example is the polymerase encoded by the HBV virus [13].

## 3. Phage DNA Polymerases in Action–DNA Replication of the Well-Studied Bacteriophages

DNA polymerases of ϕ29 and T4 bacteriophages belong to the B family, whereas phage T7 DNA polymerase is a member of the A family [9,20,21]. The replication mechanisms of three above phages differ from each other [6,7,8] and are briefly depicted in the following section.

### 3.1. DNA Replication of ϕ29 Phage

DNA polymerases require a free hydroxyl group at the 3′-end to catalyze DNA replication. Typically, it is provided by the activity of a primase, which synthesizes a short stretch of RNA on template DNA. However, the initiation of replication of bacteriophage ϕ29 does not require the presence of RNA polymerase. The bacteriophage ϕ29 has a genome in the form of double-stranded linear DNA. During infection, once the genetic material enters the host cell, the initiation of replication takes place through the p3 terminal protein covalently linked to the 5′-end of the DNA. Then the DNA ends are destabilized due to the interaction with the p6 protein [22,23,24]. It also turned out that the last and penultimate nucleotides in the genome of bacteriophage ϕ29 are TT (Figure 1A).

It was noted that the first nucleotide added is dAMP, which becomes attached to the penultimate nucleotide of the template [25,26]. It takes place with the participation of several proteins. P6 lowers the K_M_ dAMP value, thanks to which p2 catalyzes the deoxyadenylation of Ser232 in the p3 protein [23,26]. Commencement of replication at the +2 site could result in the loss of more nucleotides during subsequent replication cycles. As such phenomenon was not observed in phage ϕ29, it was proved that a sliding-back mechanism exists. It allows the dAMP-p3-p2 complex to move back to the +1 position, without shifting the template, which causes the reattachment of another nucleotide at the +2 position [26]. If the newly synthesized strand reaches a size of at least six nucleotides, DNA polymerase detaches from the p3 protein and continues replication independently. The resulting ssDNA is covered by p5 SSB [8] (Figure 1B).

### 3.2. DNA Replication of T4 Phage

The genetic material of the T4 bacteriophage is a linear dsDNA. Its replication process, as compared to the rest of those described in this paper, is more complicated. This is related to the interaction with more proteins required to carry out the process. For replication to occur, proteins encoded by phage DNA genes are required. Therefore, it is transcription that comes first. Once all the proteins are present, the first round of replication is initiated at one of several *ori* sites (*oriA*, *oriC*, *oriE*, *oriF*, or *oriG*). Depending on the place where replication initiation takes place, phage encoded transcriptional activators are required MotA for *oriA*, *oriF*, *oriG*, and DbpC for *oriE* [27]. Origins of replication sites are recognized by host RNA polymerase after replacement of the σ^70^ subunit with bacteriophage-encoded AsiA σ-factor. This results in the formation of the R loop structure on the 3′ to 5′ strand which serves as a primer for unidirectional leading-strand synthesis [28] (Figure 2A). The clamp loader gp44/gp62 complex then loads the sliding clamp gp45 at the end of the resulting heteroduplex, thereby allowing the attachment of bacteriophage gp43 DNA polymerase. The unfolding of the double-stranded construct is ensured by the gp41 helicase, which is brought to the replisome by the gp59 protein [29,30,31] (Figure 2B). In the lagging strand, the primer fragments are generated by the activity of the gp61 primase which is recruited by gp41. At a later stage, the primers are removed by RNaseH and the gaps are sealed by the gp30 ligase [6,29,32]. With each subsequent cycle, the successive replication products would be shorter and shorter. The solution to this problem is the presence of a replication mechanism by recombination [33].

Recombination-dependent replication of the bacteriophage T4 genome can follow five pathways. It depends on different conditions [34]. First, pathway I may occur when T4 DNA polymerase is inactivated by mutation. After infecting bacteria by T4 bacteriophage DNA, recombinational intermediates appear when dsDNA is fragmented and partially degraded by exonucleases and endonucleases. Second, in the absence of or impaired priming Okazaki fragments, late proteins endonuclease VII or terminase cut the invaded DNA providing primers for DNA polymerase in pathway III. Third, pathway IV is observed when T4 damaged DNA appears. It is called a double-strand-break repair (DSBR) recombination pathway. DNA synthesis is limited to repair DNA near the invasion sites and cutting occurs after formation of double-Holliday junctions. Fourth, in pathway V bubble-migration and ss-DNA annealing occur. It has been proposed to explain that asymmetries in gene conversion at intron homing sites occur because enzyme endonuclease VII is not required [34].

Pathway II is thought to play a major role in the replication of genetic material. The 3′-overhangs of lagging strands formed in this way are covered with gp32 SSB and UvsX single-strand annealing protein [35]. Such 3′-overhangs can serve as a template for DNA polymerase by attaching to the homologous strand of another DNA molecule (Figure 2C). This creates a Y structure with a D loop, while the lagging strand is synthesized from Okazaki fragments that create subsequent molecules with 3′-overhangs. For this reason, this type of DNA replication is called self-perpetuating. The end products of the reaction, i.e., the Holliday structures, are solved by gp49 endonuclease VII [34,36].

### 3.3. DNA Replication of T7 Phage

The replication of the T7 bacteriophage genome is mediated by a replication-by-transcription mechanism. Host RNA polymerase is responsible for the transcription of early genes, e.g., the gene encoding gp1 phage RNA polymerase [37]. The enzyme gp1 forms primers on the leading strands, which are then elongated by the DNA polymerase gp5 activity. Priming of the lagging strand is carried out by the bifunctional gp4 primase-helicase. Replication of the bacteriophage T7 genome is unidirectional for both strands, until the gp2.5 SSB initiates bi-directional DNA synthesis, resulting in the transformation of the replication “bubble” into a Y structure [7,38,39,40] (Figure 3A).

Primers are removed by gp6 RNaseH activity, gap filling is performed by the production of gene 5 and gap sealing by gp1.3 DNA ligase [7]. The results of the action of the above-mentioned enzymes are ssDNA 3′-overhangs. They have repeats of about 160 bp in length, allowing them to hybridize with themselves to form concatemers, which are then assembled through ligase activity (Figure 3B). During successive rounds of DNA replication, Holliday structures are formed, which are resolved by the activity of endonuclease I, encoded by gene 3. Subsequently, single-stranded 5′-overhangs are formed during concatemeric DNA packaging. The ends are filled in by DNA polymerase and exonuclease (gene 6 product), preventing strand-displacement synthesis [41,42].

## 4. Unusual Functions and Features of Recently Discovered Phage-Encoded DNA Polymerases

Below we present several recently described examples of bacteriophage-encoded DNA polymerases that display unusual, noncanonical activities and/or interesting structural features.

### 4.1. Unexpected Translesion Synthesis by B-Family Phage Polymerase from Bam35 Phage

Replicative DNA polymerases must synthesize DNA with high fidelity. That is why most of them have evolved to promote the proper base pairing; besides, the ‘proofreading’ 3′→5′ exonuclease activity of the A- and B-family polymerases additionally assures the correctness of the nascent DNA strand [43]. The cost of such an adaptation is the fact that replicative DNA polymerases cannot pass through the DNA damage, resulting in stalled replication forks and replisome disassembly [43,44]. Damage-encountering replicative polymerases are exchanged with translesion synthesis (TLS) polymerases belonging to the polymerase family Y. Due to the specific structural features of their active center, TLS polymerases are able to bypass DNA lesion, but they are highly error-prone and exhibit low processivity [45]. 

In context of phage DNA replication, high fidelity and processivity have to be balanced with damage tolerance. This problem is especially relevant in cases of temperate bacteriophages because they can accumulate DNA lesions when integrated into the host genome during the lysogenic cycle. After prophage induction, phage DNA polymerase must deal with accumulated damage during phage DNA amplification. The most frequent kind of DNA lesions occurring in bacteria are abasic (apurinic/apyrimidinic) sites caused by the spontaneous or enzymatic hydrolysis of the N-glycosidic bond between the nitrogenous base and deoxyribose sugar [46] (Figure 4A).

A phage-encoded DNA polymerase of *Bacillus thuringiensis* phage Bam35 that has a surprising ability to bypass over abasic sites during DNA replication has been described [47]. It belongs to family B and shares structural (over 40%) and functional similarity with ϕ29 DNA polymerase, along with the protein-priming mechanism. Biochemical assays in vitro showed that Bam35 polymerase exhibits efficient polymerase activity and 3′→5′ exonuclease activity. Moreover, contrary to ϕ29 DNA polymerase, wild type Bam35 proteins are able to insert a nucleotide opposite the abasic side as well as the first nucleotide beyond it, carrying on with DNA synthesis downstream of the lesion [47]. According to the so-called ‘A-rule’ used by TLS DNA polymerases [48], Bam35 polymerase preferentially incorporates dATP opposite the abasic site of DNA [47] (Figure 4B).

Protein-primed DNA polymerases contain two additional structural regions compared to the other B-family DNA polymerases. The first of them, named TPR1 (terminal protein region 1) is involved in interactions with primer terminal protein, second-TPR2–is responsible for strand displacement coupled to DNA synthesis and provides high enzyme processivity [49,50]. It has been revealed that Bam35 DNA polymerase devoid of TPR2 still can insert nucleotides opposite the abasic site but loses the bypass ability and further DNA extension. It suggests that the TPR2 subdomain in Bam35 DNA polymerase may be involved in proficient TLS activity [47].

Bam35 DNA polymerase is a unique example of fully functional replicative enzyme with additional TLS activity. The balance of damage tolerance and faithful DNA synthesis within the single protein has an application potential in the fields where amplification of damaged DNA is needed, such as criminology or paleontology [47].

### 4.2. Self-Priming Activity of Unique Hexameric Ring-Shaped DNA Polymerase from the NrS-1 Phage

It has been commonly thought for years that DNA polymerases cannot synthesize DNA de novo and require the presence of a primer attached to the 5′-end of the template. Thus, replicative DNA polymerases need the assistance of primases, specific enzymes able to synthesize short RNA fragments at specific sequences on the DNA template, that can be subsequently extended [43]. However, the discovery of the so-called primase–polymerases (prim–pols) that have a role in replicating whole plasmid sequences in archaea was proof that de novo synthesis of long DNA fragment from dNTPs is also possible [51]. Prim–pols are sometimes classified as another DNA polymerase family [52]. In fact though, members of primer-independent family B polymerases, dubbed pipolBs, encoded by self-replicating mobile genetic elements in bacterial and mitochondrial DNA, have also been identified [53].

Recently, DNA polymerase from NrS-1 bacteriophage, responsible for the replication of its genome, shed a new light on de novo DNA synthesis. NrS-1 temperate bacteriophage was described in 2013 as a first isolated phage infecting the deep-sea vent *Epsilonproteobacteria* [54]. Interestingly, despite the presence of genes coding for helicase and ss-DNA binding protein, suggesting that replication of NrS-1 phage relies on its own enzymes, NrS-1 does not encode any replicative polymerase homologous to the known ones. However, it does have a gene with a small sequence homology to the DNA prim–pols identified on archaeal plasmids. It turned out that this gene indeed encodes replicative DNA polymerase that can both extend primed DNA strands and synthesize DNA de novo. The N-terminal part of the enzyme contains prim/pol domain with homologies to archaeal prim–pols and helix bundle domain (Figure 5). The N-terminal polymerase fragment of 300 residues is responsible for the DNA polymerization depending on divalent metal ion mechanism, based solely on template strands and dNTPs [55,56]. Notably, helix bundle domain is necessary for the priming activity, because the N-terminal part of Nrs-1 polymerase containing only the first 200 residues was not able to synthesize DNA without a primer [55].

Typical primases as well as DNA prim–pols recognize their initiation site upon the 3–4 nucleotide sequence on DNA templates thanks to a zinc-binding motif in their structure [57]. In contrast, NrS-1 DNA polymerase lacks such a motif and recognizes longer, 8-nucleotide sequences via Y261 residue, which implicates that the priming mechanism is different than in already described DNA prim–pols [55,56].

**Figure 5 ijms-23-00635-f005:**
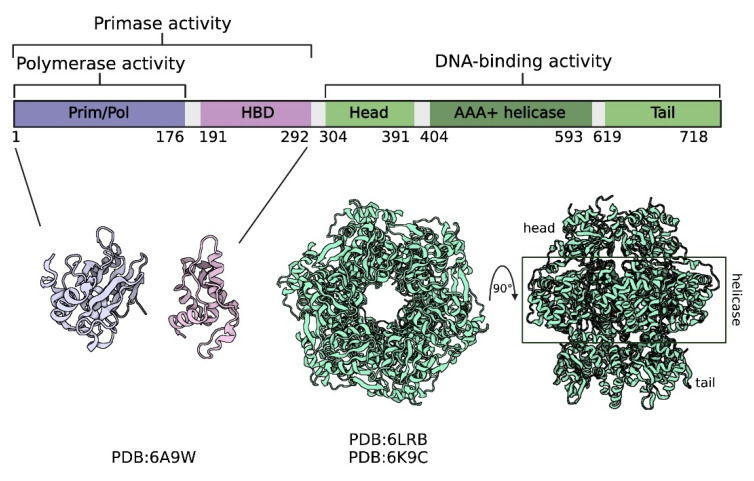
The scheme of functional domains in the NrS-1 DNA polymerase. The N-terminal part of the protein contains prim/pol domain and helix bundle domain (HBD), whereas the C-terminal region forms a hexameric ring consisting of AAA+ helicase domains and previously unknown head and tail regions (adapted from [58], with modifications).

The C-terminal part of the NrS-1 polymerase that proved to play important role in DNA binding and improving DNA polymerization efficiency has even more interesting features. Crystal structure resolution revealed that it forms a unique hexameric ring whose central region shares structural similarities with an AAA+ helicase domain, whereas head and tail residue structures have not been described before [58] (Figure 5). The C-terminal region possesses nucleotide hydrolysis and helicase activities. The latter is directly involved in enhancing the processivity of NrS-1 polymerase [58]. Electron microscopy analysis of the full-length protein revealed that the whole NrS-1 polymerase also adapts hexameric conformation unique among other DNA polymerases. Such unusual structure and function may play a role in proper replisome assembly and initiation of DNA replication during phage life cycle [58]. NrS-1 polymerase is a first example of phage main replicative enzyme with primer synthesis, DNA polymerization, and helicase activities.

### 4.3. Incorporation of Non-Canonical Nucleotides by DNA Polymerases from 2-Aminoadenine Based Phages

The DNA of all living organisms is composed of only four different nitrogenous bases: purines adenine (A), guanine (G), pyrimidines cytosine (C), and thymine (T). According to the canonical Watson–Crick rules, adenine pairs with thymine, and guanine with cytosine, forming two and three hydrogen bonds, respectively [59]. Bacteriophages, however, often use modified DNA bases to avoid recognition and degradation of their genetic material by host defense systems, i.e., restriction endonucleases. One of the flagship examples of such mechanisms is the presence of 5-hydroxymethylcytosine instead of cytosine in T-even phages, such as T2, T4, and T6 (other types of base modifications present in bacteriophage DNA and their functions are reviewed in [60]).

Recently published articles focused on especially interesting modifications of bacterial DNA in terms of DNA polymerization. Several bacteriophages that belong to the *Siphoviridae* and *Podoviridae* families, including *Vibrio* phage ϕVC8 or *Synechococcus* phage S-2L, are entirely devoid of adenine, which is replaced in their genome by 2-aminoadenine (2,6-diaminopurine, Z) [61]. Unlike an A–T pair, Z–T forms three hydrogen bonds contravening the Watson–Crick rules of base pairing [59] (Figure 6A). Genome features of S-2L phage have been already known for over 30 years [62], but the problem of its replication seems to be neglected.

S-2L and other bacteriophages whose genomes are built of aminoadenine nucleotides (referred below as ‘aminoadenine phages’) encode PurZ—an enzyme involved in the first step of dZTP synthesis from dGMP [63]. Pezo and coworkers wondered whether there exists a polymerase that specifically incorporates dZTP into DNA. After a database search, they revealed that all aminoadenine phages, except S-2L, contain a gene homolog of *polA* from *E. coli*, which they called *dpoZ* [64]. DpoZ enzymes from four different phages, overproduced in *E. coli* and purified to homogeneity, functionally resemble the Klenow fragment of family A DNA polymerase I, but show significantly higher affinity to dZTP over dATP [64]. The crystal structure of the apo form of DpoZ from vibriophage ϕVC8 showed that this polymerase is structurally quite a typical member of the A family [65]. It consists of a polymerase domain with palm, thumb, and fingers subdomains as well as a 3′→5′ exonuclease domain. Compared to *E. coli* Pol I, ϕVC8 DpoZ has three small sequence insertions that form external loops in the palm and fingers domains. However, neither single structural motif nor specific residue(s) in the polymerase domain that could justify the preference for dZTPs during DNA synthesis have been identified [65]. Therefore, substrate preference may arise from the combination of several structural features. The authors suggest that the possible mechanism of Z-vs-A specificity relates to the switching between polymerase and exonuclease activity during DNA synthesis. It was recently shown that in response to replication obstacles, DNA polymerase backtracks the nascent DNA strand and sends it to the exonuclease domain, where even a correctly incorporated nucleotide is excised [66] (Figure 6B). According to Czernecki and coworkers, backtracking activity of DpoZ enzyme could be especially high and somehow sensitive to the strength of the nascent base pair, that is to the number of forming hydrogen bonds, in favor of aminoadenine [65] (Figure 6C). Structural basics of this mechanism remain yet to be elucidated. 

Since an S-2L phage has no *dpoZ* gene homolog, the question is how it can incorporate Z over A into the DNA. It turned out that S-2L contains a gene encoding protein of the prim–pol type that seems to be its only replicative polymerase. Subsequent structural and functional analyses revealed that the S-2L DNA polymerase folds similarly to archaeal prim–pols and—surprisingly—shows no discrimination of A against Z [67]. The reason for dZTP usage in S-2L phage lies in the presence of products of two other genes, DatZ–triphosphohydrolase specific of dATP and MazZ–(d)GTP-specific diphosphohydrolase [67,68]. The former degrades dATP to remove the alternative for dZTP and the latter hydrolyzes dGTP to dGMP providing substrate for (and therefore ensuring a high rate of) dZTP synthesis by PurZ. DatZ and MazZ homologs are also found in phages encoding the DpoZ polymerase [68].

**Figure 6 ijms-23-00635-f006:**
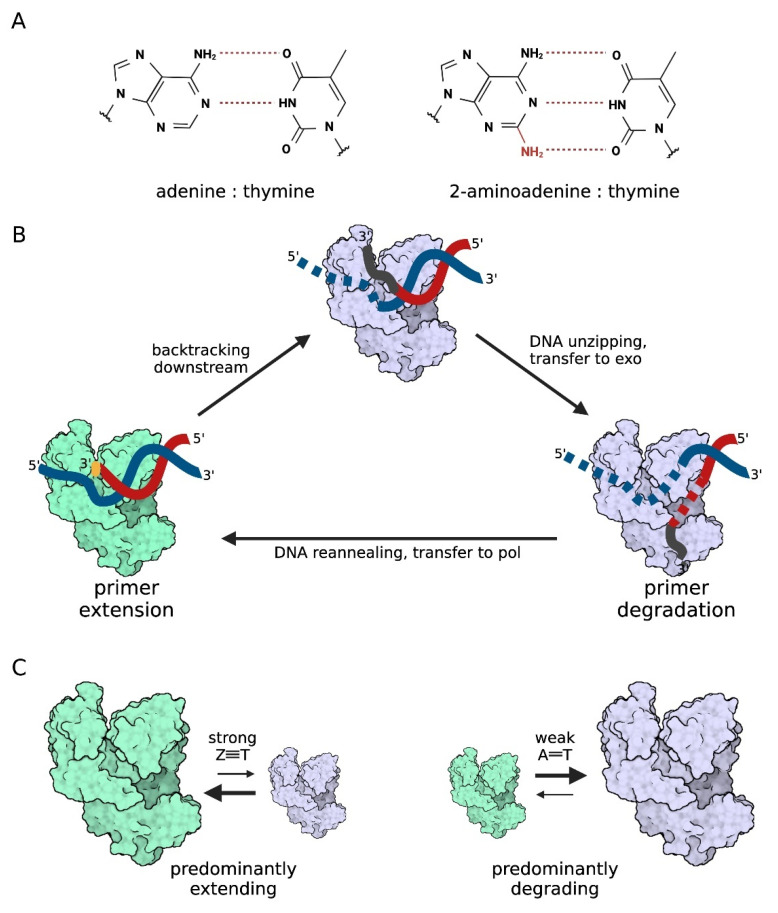
The Z-vs-A specificity of DpoZ polymerase from aminoadenine phages. (**A**) Structural formulas of adenine:thymine and 2-aminoadenine:thymine base pairs. (**B**) Mechanism of DNA polymerase backtracking and switching between polymerase and exonuclease activities. During DNA synthesis (polymerase in green), the template (blue) and primer (red) strains are tightly placed in the enzyme’s active center. The primer is extended at 3′-end (pale orange). However, the nascent strand (grey fragment) is occasionally backtracked and in consequence DNA is uncoupled and single strand fragments become disordered (dotted lines). As a result, the primer strand is sent to the exonuclease site (polymerase in light purple), where 2–3 terminal nucleotides are subsequently removed before switching to the extending mode (adapted from [65]). (**C**) DpoZ specificity may relate to high backtracking activity dependent on the number of formed hydrogen bonds. When dATP (two bonds with thymine) appears in the active site, DpoZ polymerase backtracks the nascent strand and transfers it to the exonuclease site where adenine is removed. In contrast, dZTP (three bonds with thymine) allows for the polymerase to stay predominantly in the extending mode (adapted from [65]).

Taken together, the ‘molecular ostracism’ of adenine and exclusive incorporation of dZTP into DNA result from several activities of enzymes encoded by aminoadenine phages’ genomes, at different stages of phage life cycle. At the pre-replicative stage, dATP is eliminated by the enzymatic degradation thanks to specific triphosphohydrolase while concomitantly dZTP synthesis is promoted by providing high amounts of dGMP. At replicative stage, in turn, using dZTP as a DNA building block is supported by newly described features of DpoZ DNA polymerase [64,65,67].

## 5. Bacteriophage-Encoded DNA Polymerases as a Powerful Application Tool in Biotechnology

There are few DNA polymerases encoded by bacteriophages that have found applications in biotechnology. This is not due to a lack of applications in this field, but rather to limited knowledge about them. However, ongoing research makes it possible to learn more and more about the newly discovered enzymes encoded by these viruses. The most known enzymes that are used in genetic engineering are DNA polymerases encoded by bacteriophages φ29, T4 and T7.

Several applications of DNA polymerases of T4 and T7 phages can be found in the literature. Due to their similar activities (5′ to 3′ polymerase activity, 3′ to 5′ exonuclease activity) and properties, they also have similar fields of application. Both can be effectively used in filling 5′-overhangs or removing 3′-overhangs [69]. Another common application is DNA strand elongation during site-specific mutagenesis [70,71]. However, despite these similarities, the aforementioned DNA polymerases obviously differ in applications. The DNA polymerase encoded by phage T4 can be used during the synthesis of labelled DNA by replacement reaction and in cloning independent of ligation of PCR products [72]. In contrast, phage T7 DNA polymerase is used to remove residual genomic DNA during circular DNA purification. A major advantage of T7 DNA polymerase is the efficient elongation of templates with a large number of base pairs. It is also used to synthesize complementary cDNA strands [73]. Another use is the possibility of labelling the 3′ ends of DNA [73]. An interesting application is its use for the in situ detection of fragmented DNA as a result of apoptosis, thanks to its ability to label niches present in double-stranded DNA [74].

Among all bacteriophage-encoded DNA polymerases, φ29 DNA polymerase (φ29 DNAP) has significantly widespread application potential for several reasons. Apart from the protein-priming mechanism described before, φ29 DNAP also accepts DNA- or RNA-primers. Its high fidelity due to 3′ to 5′ proofreading activity and extremely high processivity accompanied with strand displacement activity [75] makes it a useful enzyme for various isothermal DNA amplification methods. The most popular ones where φ29 DNAP is used are rolling circle amplification [76] and multiple displacement amplification [77], which is now commonly used for the whole genome amplification [78]. Moreover, it has been recently shown that φ29 DNAP exhibits limited reverse transcriptase activity that enables it to replicate circular, RNA-containing templates, and thus broadens the application opportunities [79]. The construction of chimeric φ29 DNAP with enhanced DNA binding due to the fusion with helix–hairpin–helix domains [80] and in vitro evolution of φ29 DNAP performed to date [81] clearly show that this enzyme is quite engineering-prone and even further improving of its activity is possible. 

The DNA polymerase of phage φ29 has also found its application in DNA sequencing. For instance, it has been successfully used in single molecule real-time sequencing where immobilized DNA polymerase incorporates labelled nucleotides that are fluorescently detected [82,83]. An especially interesting example of φ29 DNAP application is nanopore sequencing. In this technique, a pore protein is inserted into a membrane separating two salt solutions. When a voltage is applied across the membrane, a nascent DNA strand synthesized by polymerase is passed through the pore and this DNA movement causes the sequence-dependent change of ionic current. Manrao and coworkers combined modified *Mycobacterium smegmatis* porin A protein MspA with highly processive φ29 DNAP. While synthesizing DNA, φ29 DNAP functions as a motor that pulls single-stranded DNA through the MspA protein and detects current changes that clearly correspond to the known DNA sequences [84]. The same system turned out to be effective to detect unnatural bases in DNA, such as dNaM and d5SICS. Here, φ29 DNA polymerase is flexible enough to unzip DNA strands containing unnatural bases and move them through the MspA where base-specific current levels can be detected [85]. 

Unexpectedly high flexibility of φ29 DNAP allows for the use of this enzyme for xenobiotic nucleic acids (XNAs) biosynthesis. XNAs are synthetic genetic polymers that have at least one chemical moiety (phosphate, sugar, or nucleobase) changed in comparison to their natural equivalents. The major advantage of XNAs over DNA or RNA is their higher chemical and biological stability, especially useful in the construction of novel nanostructures [86]. Natural DNA polymerases do not generally manage to accept backbone modifications in XNAs, and they require extensive engineering to cope with XNAs synthesis. However, it has been revealed that only one amino acid substitution impairing exonuclease activity in φ29 DNAP makes this enzyme able to synthesize XNAs [87]. This polymerase has been shown to be an effective enzyme for the synthesis of 1,5-anhydrohexitol nucleic acid (HNA), 2′-deoxy-2′-fluoro-arabinonucleic acid (FANA), and 2′-fluoro-DNA, where the canonical ribofuranose ring is replaced with hexitol, fluoroarabinose, and 2′-fluoro-2′-deoxyribose, respectively [86,87]. Because only a single mutation is sufficient to provide XNAs synthesis (whereas 14 substitutions have been required to obtain the same activity of archaeal DNA polymerase from *Thermococcus gorgonarius* [87]), highly processive φ29 DNAP (and possibly other bacteriophage-encoded DNA polymerases) constitutes a significant perspective for further XNAs research and development of XNA-based biomaterials.

## 6. Concluding Remarks

Bacteriophages are the most numerous and widespread biological entities in the world [88]. They not only have adapted to the environment of their hosts but also developed numerous mechanisms to escape hosts’ defense systems. That is why DNA polymerases encoded in phages’ genomes may exhibit features and activities that have not been observed before.

Examples of recent discoveries presented in this article prove that phage-encoded DNA polymerases can show activities unusual for the members of the family they belong to [47], display unique, previously undescribed structures [58], or even disobey canonical rules of base pairing during DNA synthesis adopted by all living organisms [64].

Even in single habitat the biodiversity of bacteriophages is enormous [89]. However, the number of complete phage genome sequences deposited in genetic databases is relatively small (10,830 of complete phage genomes, in comparison to almost 364,000 bacterial genomes, deposited in GenBank, as accessed on 30 November 2021; https://www.ncbi.nlm.nih.gov/genome/browse/#!/viruses/), especially when one considers their biological abundance. That is why newly isolated bacteriophages from environmental samples need to be extensively characterized in search of genes encoding unknown proteins or enzymes of unusual activities. For instance, the analysis of a group of bacteriophages isolated from urban sewage revealed that three of them possessed genes encoding for DNA polymerases [90,91]. DNA polymerase from *Enterococcus faecalis*-infecting bacteriophage vB_EfaS-271 is predicted to belong to family B [90], but it does not show high similarity to other members of this family. Therefore, studies on DNA polymerases from newly described bacteriophages would provide better understanding of their action during the phage life cycle, which could also be useful in terms of the potential application of phage polymerases in biotechnology.

## Figures and Tables

**Figure 1 ijms-23-00635-f001:**
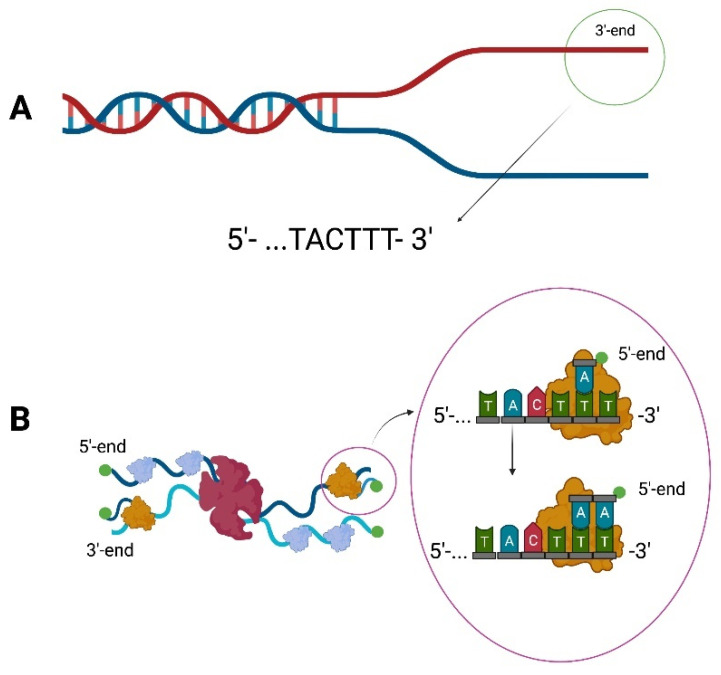
Replication of the bacteriophage ϕ29 genome. (**A**) The 3′-terminal region of bacteriophage ϕ29 genome. (**B**) The p6 protein (red) destabilizes the ends of the DNA, making replication possible. The initiation of replication of bacteriophage ϕ29 occurs through the terminal protein p3 (green) which is covalently bound to the 5′ end. DNA polymerase (orange) in complex with p3 protein and dAMP pairs the first nucleotide with the penultimate T from the template. A sliding-back mechanism then occurs, which results in the entire complex moving to the +1 position of the newly synthesized strand, allowing re-matching at the penultimate template site. The ssDNA is covered by p5 SSB (blue). Schemes follow the same formatting.

**Figure 2 ijms-23-00635-f002:**
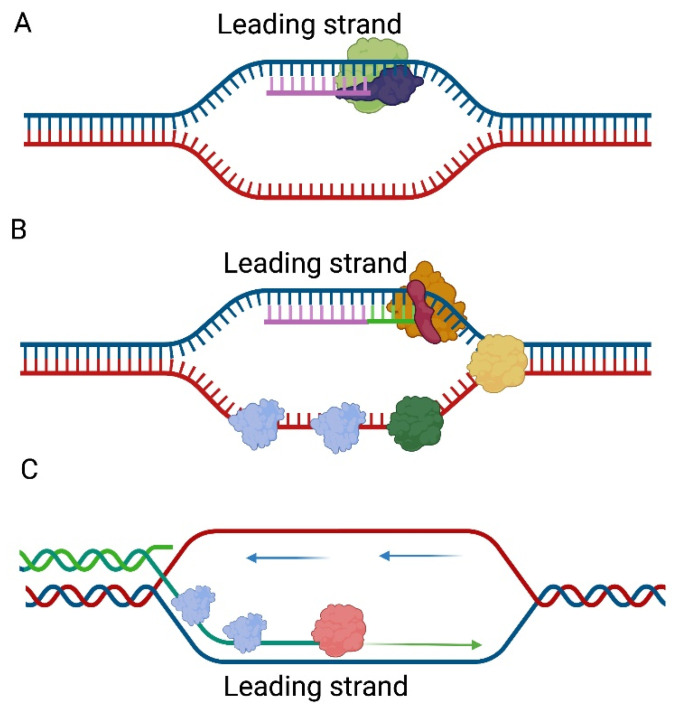
Replication of the bacteriophage T4 genome. (**A**) Origin of replication sites are recognized by host RNA polymerase (green) after replacement of the σ^70^ subunit with bacteriophage-encoded AsiA σ-factor (blue). (**B**) The DNA polymerase gp43 (orange) needs sliding clamp gp45 (red) for replication of DNA. The unfolding of the double-stranded construct is ensured by the gp41 helicase (yellow). In the lagging strand, the primer fragments are generated by the activity of the gp61 primase (green) which is recruited by gp41 (yellow). The ssDNA structures are covered with gp32 SSB protein (blue). (**C**) After removal of RNA primers, the 3′-overhangs of lagging strand are covered with gp32 SSB (blue) and UvsX single-strand annealing protein (pink). Such a 3′-overhangs (green) can serve as a template for DNA polymerase by attaching to the homologous strand of another DNA molecule (blue and red). This creates a Y structure with a D loop, while the lagging strand is synthesized from Okazaki fragments that create subsequent molecules with 3′-overhangs.

**Figure 3 ijms-23-00635-f003:**
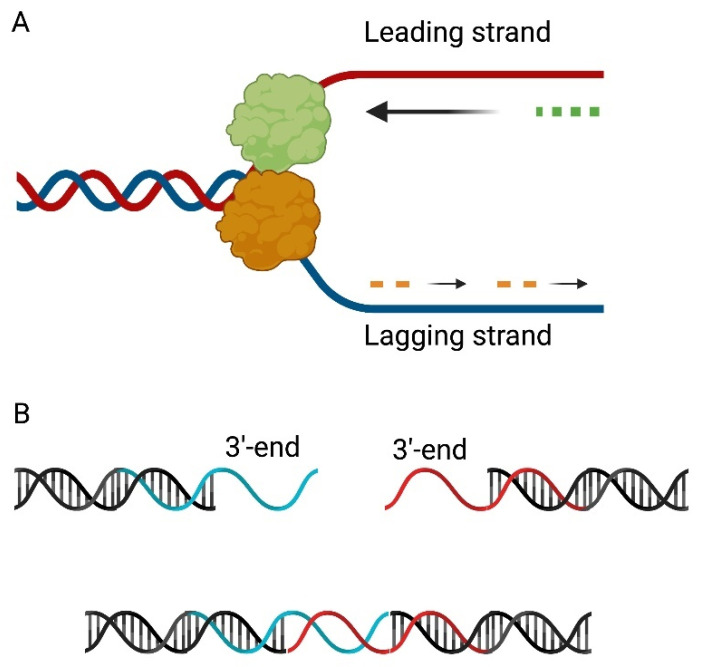
Replication of the bacteriophage T7 genome. (**A**) RNA polymerase, encoded by gene 1, recognizes the origins of replication and synthesizes the transcript (green dots). After initiation of bidirectional DNA synthesis by a 2.5 SSB gene product, the replication “bubble” transforms into a Y-structure. Priming (orange dots) of the lagging strand is carried out by the bifunctional protein primase-helicase (orange). (**B**) After removal of RNA primers, 3′-overhangs are formed which have repeats of about 160bp in length, allowing it to hybridize with itself to form concatemers.

**Figure 4 ijms-23-00635-f004:**
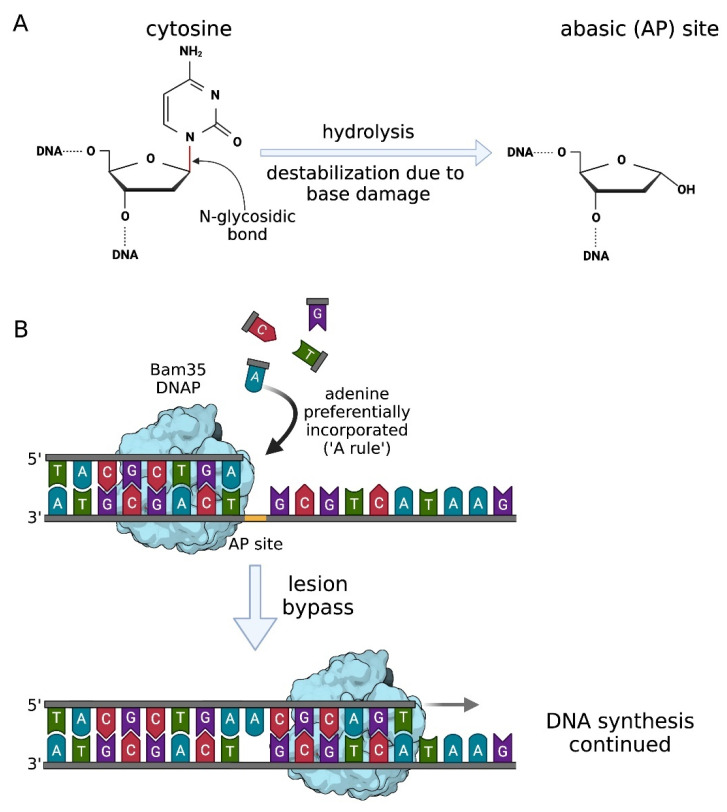
Abasic site bypass via the A-rule and translesion synthesis by Bam35 DNA polymerase. (**A**) Abasic AP sites form after the cleavage of the N-glycosidic bond between nitrogenous base and deoxyribose sugar, whereas the phosphodiester backbone of DNA remains intact. Base loss may occur due to spontaneous or enzymatic hydrolysis as well as base damage leading to destabilization of N-glycosidic bond (adapted from [46], with modifications). (**B**) Translesion synthesis by Bam35 DNA polymerase. Once encountering an abasic site, Bam35DNAP (light blue) incorporates dATP opposite the lesion, passes through it and continues DNA synthesis in a faithful manner [47].

**Table 1 ijms-23-00635-t001:** Families of DNA polymerases.

DNA Polymerase Family	Taxa	Example(s)	Main Function(s)
A	Eukaryota	Pol γ, Pol θ, Pol v	Replication, repair
	Bacteria	Pol I	
	Viruses	T7 DNA pol	
B	Eukaryota	Pol ζ, α, σ	Replication, repair
	Bacteria	Pol II	
	Archea	DNA pol B	
	Viruses	T4 DNA pol	
C	Bacteria	Pol III	Replication
D	Archea	Pol D	Replication
X	Eukaryota	Pol β, σ, λ, μ	Repair
	Bacteria	Pol X	
	Archea	Pol X	
	Viruses	ASFV DNA pol	
Y	Eukaryota	Pol ι, κ, η	Translesion synthesis
	Bacteria	Pol IV i V	
	Archea	Dpo4 DNA pol	
RT	Eukaryota	Telomerase	RNA-dependent DNA synthesis
	Viruses	Pol HBV virus	

## Data Availability

Not applicable.

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
