# Peer review of "Bacteriophage-Encoded DNA Polymerases—Beyond the Traditional View of Polymerase Activities"

_ijms, 2022, doi:10.3390/ijms23020635_

Round 1
Reviewer 1 Report
The review entitled “Bacteriophage-encoded DNA polymerases – beyond the traditional view of polymerase activities” contains information on different features of DNA polymerases encoded by numerous bacteriophages. Additionally, the Authors explained the protein-priming machinery to initiate DNA synthesis by phage enzymes. The Authors have thoroughly analyzed the existing literature and specifically mentioned the related points with proper explanations. However, a few points are to be taken into consideration to enrich with more factual information in the line of the topic. My comments have been listed below:
Major comments:
- On page-3, Line no-111-112. “DNA polymerases of ϕ29 and T4 belong to the B family, whereas T7 DNA polymerase is a member of the family A”. Please write “bacteriophages” after T4.
- On page-5-6, Line no-180-200, “Recombination-dependent replication of the bacteriophage T4……………..”. Can the Authors put some points on R-loop i.e. RNA-DNA hybrid formation after infection. Please also mention that annealed RNA at replication origin serves as a primer for one-directional leading-strand synthesis.
- The Authors are encouraged to write future implications with therapeutics and applications in genetic engineering and biotechnology.
Minor comments:
- On page-2, Line no-60. It should be “5’ to 3’ DNA synthesis activity”
- In Reference 23, the Journal’s name is missing. Please add it.
- Please thoroughly check the whole manuscript again.
Author Response
Major comments:
1. On page-3, Line no-111-112. “DNA polymerases of ϕ29 and T4 belong to the B family, whereas T7 DNA polymerase is a member of the family A”. Please write “bacteriophages” after T4.
ANSWER: Modified as requested by the reviewer (lines 111-112)
2. On page-5-6, Line no-180-200, “Recombination-dependent replication of the bacteriophage T4……………..”. Can the Authors put some points on R-loop i.e. RNA-DNA hybrid formation after infection. Please also mention that annealed RNA at replication origin serves as a primer for one-directional leading-strand synthesis.
ANSWER: This is mentioned now, as suggested by the reviewer (lines 156-158)
3. The Authors are encouraged to write future implications with therapeutics and applications in genetic engineering and biotechnology.
ANSWER: As suggested, a new chapter on applications of the polymerases in genetic engineering and biotechnology has been added (lines 416-485).
Minor comments:
1. On page-2, Line no-60. It should be “5’ to 3’ DNA synthesis activity”
ANSWER: Corrected (line 60)
2. In Reference 23, the Journal’s name is missing. Please add it.
ANSWER: Journal name has been added (Ref. 23)
3. Please thoroughly check the whole manuscript again.
ANSWER: As suggested, we have checked the whole manuscript carefully and corrected remaining minor errors.
Reviewer 2 Report
The manuscript entitled Bacteriophage-encoded DNA polymerases beyond the traditional view of polymerase activities by Morcinek-Orlowska et al., summarizes the properties of a few bacteriophage DNA polymerases. The manuscript is well written and informative. The figures are very clear and helpful for the understanging. It could be published as it is but it would benefit for mentioning several applications. For example, Phi 29 polymerase which is extremely processive can be used for nanopore sequencing( Nature Biotech 2012). Phi 29 can also be used for the detection of non-natural bases such as dNaM and d5SICS (JACS). Phi29 DNAP is an efficient catalyst for the production of various artificial nucleic acids (XNAs) carrying backbone modifications such as 1,5-anhydrohexitol nucleic acid (HNA), 2'-deoxy-2'-fluoro-arabinonucleic acid (FANA), and 2'-fluoro-2'-deoxyribonucleic acid (2'-fluoro-DNA).
There are several typing errors
lane 362 polA should be italicized
lanes363 and 364 E. coli and polA should be italicized
lane 405 site should replaced side
lane 382 dpoZ should be italicized
Author Response
REVIEWER'S COMMENT:
The manuscript entitled Bacteriophage-encoded DNA polymerases beyond the traditional view of polymerase activities by Morcinek-Orlowska et al., summarizes the properties of a few bacteriophage DNA polymerases. The manuscript is well written and informative. The figures are very clear and helpful for the understanging. It could be published as it is but it would benefit for mentioning several applications. For example, Phi 29 polymerase which is extremely processive can be used for nanopore sequencing( Nature Biotech 2012). Phi 29 can also be used for the detection of non-natural bases such as dNaM and d5SICS (JACS). Phi29 DNAP is an efficient catalyst for the production of various artificial nucleic acids (XNAs) carrying backbone modifications such as 1,5-anhydrohexitol nucleic acid (HNA), 2'-deoxy-2'-fluoro-arabinonucleic acid (FANA), and 2'-fluoro-2'-deoxyribonucleic acid (2'-fluoro-DNA).
ANSWER: A new chapter on applications of the polymerases in genetic engineering and biotechnology has been added (lines 417-486), as suggested by the reviewer.
REVIEWER'S COMMENT:
There are several typing errors
lane 362 polA should be italicized
lanes363 and 364 E. coli and polA should be italicized
lane 405 site should replaced side
lane 382 dpoZ should be italicized
ANSWER:
As the reviewer indictated several typographical errors, we have checked the whole manuscript carefully and corrected the mentioned and all remaining minor typographical errors.
Round 2
Reviewer 1 Report
The Authors thoroughly checked and revised the manuscript. All queries are addressed point-wise in the revised manuscript.